# Neural Universal Discrete Denoiser

**Taesup Moon**
DGIST
Daegu, Korea 42988
tsmoon@dgist.ac.kr

**Seonwoo Min, Byunghan Lee, Sungroh Yoon**
Seoul National University
Seoul, Korea 08826
{mswzeus, styxkr, sryoon}@snu.ac.kr

## Abstract

We present a new framework of applying deep neural networks (DNN) to devise a universal discrete denoiser. Unlike other approaches that utilize supervised learning for denoising, we do not require any additional training data. In such setting, while the ground-truth label, *i.e.*, the clean data, is not available, we devise "pseudo-labels" and a novel objective function such that DNN can be trained in a same way as supervised learning to become a discrete denoiser. We experimentally show that our resulting algorithm, dubbed as Neural DUDE, significantly outperforms the previous state-of-the-art in several applications with a systematic rule of choosing the hyperparameter, which is an attractive feature in practice.

## 1 Introduction

Cleaning noise-corrupted data, *i.e.*, denoising, is a ubiquotous problem in signal processing and machine learning. Discrete denoising, in particular, focuses on the cases in which both the underlying clean and noisy data take their values in some finite set. Such setting covers several applications in different domains, such as image denoising [1, 2], DNA sequence denoising [3], and channel decoding [4].

A conventional approach for addressing the denoising problem is the Bayesian approach, which can often yield a computationally efficient algorithm with reasonable performance. However, limitations can arise when the assumed stochastic models do not accurately reflect the real data distribution. Particularly, while the models for the noise can often be obtained relatively reliably, obtaining the accurate model for the original clean data is more tricky; the model for the clean data may be wrong, changing, or may not exist at all.

In order to alleviate the above mentioned limitations, [5] proposed a universal approach for discrete denoising. Namely, they first considered a general setting that the clean finite-valued source symbols are corrupted by a discrete memoryless channel (DMC), a noise mechanism that corrupts each source symbol independently and statistically identically. Then, they devised an algorithm called DUDE (Discrete Universal DEnoiser) and showed rigorous performance guarantees for the *semi-stochastic setting*; namely, that where no stochastic modeling assumptions are made on the underlying source data, while the corruption mechanism is assumed to be governed by a *known* DMC. DUDE is shown to *universally* attain the optimum denoising performance for any source data as the data size grows.

In addition to the strong theoretical performance guarantee, DUDE can be implemented as a computationally efficient sliding window denoiser; hence, it has been successfully applied and extended to some practical applications, *e.g.*, [1, 3, 4, 2]. However, it also had limitations; namely, the performance is sensitive on the choice of sliding window size $k$, which has to be hand-tuned without any systematic rule. Moreover, when $k$ becomes large and the alphabet size of the signal increases, DUDE suffers from the data sparsity problem, which significantly deteriorates the performance.

In this paper, we present a novel framework of addressing above limitations of DUDE by adopting the machineries of deep neural networks (DNN) [6], which recently have seen great empirical success

in many practical applications. While there have been some previous attempts of applying neural networks to grayscale image denoising [7, 8], they all remained in supervised learning setting, *i.e.*, large-scale training data that consists of clean and noisy image pairs was necessary. Such approach requires significant computation resources and training time and is not always transferable to other denoising applications, in which collecting massive training data is often expensive, *e.g.*, DNA sequence denoising [9].

Henceforth, we stick to the setting of DUDE, which requires no additional data other than the given noisy data. In this case, however, it is not straightforward to adopt DNN since there is no ground-truth label for supervised training of the networks. Namely, the target label that a denoising algorithm is trying to estimate from the observation is the underlying clean signal, hence, it can never be observed to the algorithm. Therefore, we carefully exploit the known DMC assumption and the finiteness of the data values, and devise "pseudo-labels" for training DNN. They are based on the *unbiased estimate* of the true loss a denoising algorithm is incurring, and we show that it is possible to train a DNN as a universal discrete denoiser using the devised pseudo-labels and generalized cross-entropy objective function. As a by-product, we also obtain an accurate estimator of the true denoising performance, with which we can systematically choose the appropriate window size $k$. In results, we experimentally verify that our DNN based denoiser, dubbed as Neural DUDE, can achieve significantly better performance than DUDE maintaining robustness with respect to $k$. Furthermore, we note that although the work in this paper is focused on discrete denoising, we believe the proposed framework can be extended to the denoising of continuous-valued signal as well, and we defer it to the future work.

## 2    Notations and related work

### 2.1    Problem setting of discrete denoising

Throughout this paper, we will generally denote a sequence ($n$-tuple) as, *e.g.*, $a^n = (a_1, \ldots, a_n)$, and $a_i^j$ refers to the subsequence $(a_i, \ldots, a_j)$. In discrete denoising problem, we denote the clean, underlying source data as $x^n$ and assume each component $x_i$ takes a value in some finite set $\mathcal{X}$. The source sequence is corrupted by a DMC and results in a noisy version of the source $z^n$, of which each component $z_i$ takes a value in , again, some finite set $\mathcal{Z}$. The DMC is completely characterized by the channel transition matrix $\mathbf{\Pi} \in \mathbb{R}^{|\mathcal{X}| \times |\mathcal{Z}|}$, of which the $(x, z)$-th element, $\mathbf{\Pi}(x, z)$, stands for $\Pr(Z_i = z | X_i = x)$, *i.e.*, the conditional probability of the noisy symbol taking value $z$ given the original source symbol was $x$. An essential but natural assumption we make is that $\mathbf{\Pi}$ is of the *full row rank*.

Upon observing the entire noisy data $z^n$, a discrete denoiser reconstructs the original data with $\hat{X}^n = (\hat{X}_1(z^n), \ldots, \hat{X}_n(z^n))$, where each reconstructed symbol $\hat{X}_i(z^n)$ also takes its value in a finite set $\hat{\mathcal{X}}$. The goodness of the reconstruction by a discrete denoiser $\hat{X}^n$ is measured by the average loss, $L_{\hat{X}^n}(X^n, Z^n) = \frac{1}{n} \sum_{i=1}^{n} \Lambda(x_i, \hat{X}_i(z^n))$, where $\Lambda(x_i, \hat{x}_i)$ is a single-letter loss function that measures the loss incurred by estimating $x_i$ with $\hat{x}_i$ at location $i$. The loss function can be also represented with a loss matrix $\mathbf{\Lambda} \in \mathbb{R}^{|\mathcal{X}| \times |\hat{\mathcal{X}}|}$. Throughout the paper, for simplicity, we will assume $\mathcal{X} = \mathcal{Z} = \hat{\mathcal{X}}$, thus, assume that $\mathbf{\Pi}$ is invertible.

### 2.2    Discrete Universal DEnoiser (DUDE)

DUDE in [5] is a two-pass algorithm that has a linear complexity in the data size $n$. During the first pass, the algorithm with the window size $k$ collects the statistics vector

$$\mathbf{m}[z^n, l^k, r^k](a) = \big| \{ i : k + 1 \leq i \leq n - k, z_{i-k}^{i+k} = l^k a r^k \} \big|, \tag{1}$$

for all $a \in \mathcal{Z}$, which is the count of the occurrence of the symbol $a \in \mathcal{Z}$ along the noisy sequence $z^n$ that has the *double-sided context* $(l^k, r^k) \in \mathcal{Z}^{2k}$. Once the $\mathbf{m}$ vector is collected, for the second pass, DUDE applies the rule

$$\hat{X}_{i,\text{DUDE}}(z^n) = \arg \min_{\hat{x} \in \mathcal{X}} \mathbf{m}[z^n, \mathbf{c}_i]^{\top} \mathbf{\Pi}^{-1} [\lambda_{\hat{x}} \odot \pi_{z_i}] \ \text{ for each } k + 1 \leq i \leq n - k, \tag{2}$$

where $\mathbf{c}_i \triangleq (z_{i-k}^{i-1}, z_{i+1}^{i+k})$ is the context of $z_i$, $\pi_{z_i}$ is the $z_i$-th column of the channel matrix $\mathbf{\Pi}$, $\lambda_{\hat{x}}$ is the $\hat{x}$-th column of the loss matrix $\mathbf{\Lambda}$, and $\odot$ stands for the element-wise product. The form of (2)

shows that DUDE is a *sliding window denoiser* with window size $2k + 1$; namely, DUDE returns the same denoised symbol at all locations $i$'s with the same value of $z_{i-k}^{i+k}$. We will call such denoisers as the $k$-th order sliding window denoiser from now on.

DUDE is shown to be universal, *i.e.*, for *any* underlying clean sequence $x^n$, it can always attain the performance of the *best* $k$-th order sliding window denoiser as long as $k|\mathcal{Z}|^{2k} = o(n/\log n)$ holds [5, Theorem 2]. For more rigorous analyses, we refer to the original paper [5].

## 2.3   Deep neural networks (DNN) and related work

Deep neural networks (DNN), often dubbed as deep learning algorithms, have recently made significant impacts in several practical applications, such as speech recognition, image recognition, and machine translation, etc. For a thorough review on recent progresses of DNN, we refer the readers to [6] and refereces therein.

Regarding denoising, [7, 8, 10] have successfully applied the DNN to grayscale image denoising by utilizing supervised learning at the small image patch level. Namely, they generated clean and noisy image patches and trained neural networks to learn a mapping from noisy to clean patches. While such approach attained the state-of-the-art performance, as mentioned in Introduction, it has several limitations. That is, it typically requires massive amount of training data, and multiple copies of the data need to be generated for different noise types and levels to achieve robust performance. Such requirement of large training data cannot be always met in other applications, *e.g.*, in DNA sequence denoising, collecting large scale clean DNA sequences is much more expensive than obtaining training images on the web. Moreover, for image denoising, working in the small patch level makes sense since the image patches may share some textual regularities, but in other applications, the characterstics of the given data for denoising could differ from those in the pre-collected training set. For instance, the characteristics of substrings of DNA sequences vary much across different species and genes, hence, the universal setting makes more sense in DNA sequence denoising.

## 3   An alternative interpretation of DUDE

### 3.1   Unbiased estimated loss

In order to make an alternative interpretation of DUDE, which can be also found in [11], we need the tool developed in [12]. To be self-contained, we recap the idea here. Consider a single letter case, namely, a clean symbol $x$ is corrupted by $\mathbf{\Pi}$ and resulted in the noisy observation[1] $Z$. Then, suppose a single-symbol denoiser $s : \mathcal{Z} \to \hat{\mathcal{X}}$ is applied and obtained the denoised symbol $\hat{X} = s(Z)$. In this case, the true loss incurred by $s$ for the clean symbol $x$ and the noisy observation $Z$ is $\Lambda(x, s(Z))$. It is clear that $s$ cannot evaluate its loss since it does not know what $x$ is, but the following shows an unbiased estimate of the expected true loss, which is only based on $Z$ and $s$, can be derived.

First, denote $\mathcal{S}$ as the set of all possible single-symbol denoisers. Note $|\mathcal{S}| = |\hat{\mathcal{X}}|^{|\mathcal{Z}|}$. Then, we define a matrix $\boldsymbol{\rho} \in \mathbb{R}^{|\mathcal{X}| \times |\mathcal{S}|}$ with

$$\boldsymbol{\rho}(x, s) = \sum_{z \in \mathcal{Z}} \Pi(x, z)\Lambda(x, s(z)) = \mathbb{E}_x \Lambda(x, s(Z)), \quad x \in \mathcal{X}, s \in \mathcal{S}. \tag{3}$$

Then, we can define an estimated loss matrix[2] $\mathbf{L} \triangleq \mathbf{\Pi}^{-1}\boldsymbol{\rho} \in \mathbb{R}^{|\mathcal{Z}| \times |\mathcal{S}|}$. With this definition, we can show that $\mathbf{L}(Z, s)$ is an unbiased estimate of $\mathbb{E}_x\Lambda(x, s(Z))$ as follows (as shown in [12]):

$$\mathbb{E}_x \mathbf{L}(Z, s) = \sum_z \mathbf{\Pi}(x, z) \sum_{x'} \mathbf{\Pi}^{-1}(z, x')\boldsymbol{\rho}(x', s) = \delta(x, x')\boldsymbol{\rho}(x', s) = \boldsymbol{\rho}(x, s) = \mathbb{E}_x\Lambda(x, s(Z)).$$

### 3.2   DUDE: Minimizing the sum of estimated losses

As mentioned in Section 2.2, DUDE with context size $k$ is the $k$-th order sliding window denoiser. Generally, we can denote such $k$-th order sliding window denoiser as $s_k : \mathcal{Z}^{2k+1} \to \hat{\mathcal{X}}$, which

obtains the reconstruction at the $i$-th location as

$$\hat{X}_i(z^n) = s_k(z_{i-k}^{i+k}) = s_k(\mathbf{c}_i, z_i). \tag{4}$$

To recall, $\mathbf{c}_i = (z_{i-k}^{i-1}, z_{i+1}^{i+k})$. Now, from the formulation (4), we can interpret that $s_k$ defines a single-symbol denoiser at location $i$, *i.e.*, $s_k(\mathbf{c}_i, \cdot)$, depending on $\mathbf{c}_i$. With this view on $s_k$, as derived in [11], we can show that the DUDE defined in (2) is equivalent to finding a single-symbol denoiser

$$s_{k,\text{DUDE}}(\mathbf{c}, \cdot) = \arg\min_{s \in \mathcal{S}} \sum_{\{i : \mathbf{c}_i = \mathbf{c}\}} \mathbf{L}(z_i, s), \tag{5}$$

for each context $\mathbf{c} \in \mathbf{C}_k \triangleq \{(l^k, r^k) : (l^k, r^k) \in \mathcal{Z}^{2k}\}$ and obtaining the reconstruction at location $i$ as $\hat{X}_{i,\text{DUDE}}(z^n) = s_{k,\text{DUDE}}(\mathbf{c}_i, z_i)$. The interpretation (5) gives some intuition on why DUDE enjoys strong theoretical guarantees in [5]; since $\mathbf{L}(Z_i, s)$ is an unbiased estimate of $\mathbb{E}_{x_i} \Lambda(x_i, s(Z_i))$, $\sum_{i \in \{i : \mathbf{c}_i = \mathbf{c}\}} \mathbf{L}(Z_i, s)$ will concentrate on $\sum_{i \in \{i : \mathbf{c}_i = \mathbf{c}\}} \Lambda(x_i, s(Z_i))$ as long as $|\{i : \mathbf{c}_i = \mathbf{c}\}|$ is sufficiently large. Hence, the single symbol denoiser that minimizes the sum of the estimated losses for each $\mathbf{c}$ (*i.e.*, (5)) will also make the sum of the true losses small, which is the goal of a denoiser.

We can also express (5) using vector notations, which will become useful for deriving the Neural DUDE in the next section. That is, we let $\Delta^{|\mathcal{S}|}$ be a probability simplex in $\mathbb{R}^{|\mathcal{S}|}$. (Suppose we have uniquely assigned each coordinate of $\mathbb{R}^{|\mathcal{S}|}$ to each single-symbol denoiser in $\mathcal{S}$ from now on.) Then, we can define a probability vector for each $\mathbf{c}$,

$$\hat{\mathbf{p}}(\mathbf{c}) \triangleq \arg\min_{\mathbf{p} \in \Delta^{|\mathcal{S}|}} \Big( \sum_{\{i : \mathbf{c}_i = \mathbf{c}\}} \mathbb{1}_{z_i}^\top \mathbf{L} \Big) \mathbf{p}, \tag{6}$$

which will be on the vertex of $\Delta^{|\mathcal{S}|}$ that corresponds to $s_{k,\text{DUDE}}(\mathbf{c}, \cdot)$ in (5). The reason is because the objective function in (6) is a linear function in $\mathbf{p}$. Hence, we can simply obtain $s_{k,\text{DUDE}}(\mathbf{c}, \cdot) = \arg\max_s \hat{\mathbf{p}}(\mathbf{c})_s$, where $\hat{\mathbf{p}}(\mathbf{c})_s$ stands for the $s$-th coordinate of $\hat{\mathbf{p}}(\mathbf{c})$.

## 4 Neural DUDE: A DNN based discrete denoiser

As seen in the previous section, DUDE utilizes the estimated loss matrix $\mathbf{L}$, which does not depend on the clean sequence $x^n$. However, the main drawback of DUDE is that, as can be seen in (5), it treats each context $\mathbf{c}$ independently from others. Namely, when the context size $k$ grows, then the number of different contexts $|\mathbf{C}_k| = |\mathcal{Z}|^{2k}$ will grow exponentially with $k$, hence, the sample size for each context $|\{i : \mathbf{c}_i = \mathbf{c}\}|$ will decrease exponentially for a given sequence length $n$. Such phenomenon will hinder the concentration of $\sum_{i \in \{i : \mathbf{c}_i = \mathbf{c}\}} \mathbf{L}(Z_i, s)$ mentioned in the previous section, which causes the performance of DUDE deteriorate when $k$ grows too large.

In order to resolve above problem, we develop Neural DUDE, which adopts a *single* neural network such that the information from similar contexts can be shared via network parameters. We note that our usage of DNN resembles that of the neural language model (NLM) [13], which improved upon the conventional $N$-gram models. The difference is that NLM is essentially a prediction problem, hence the ground truth label for supervised training is easily availble, but in denoising, this is not the case. Before describing the algorithm more in detail, we need one following lemma.

### 4.1 A lemma

Let $\mathbb{R}_+^{|\mathcal{S}|}$ be the space of all $|\mathcal{S}|$-dimensional vectors of which elements are nonnegative. Then, for any $\mathbf{g} \in \mathbb{R}_+^{|\mathcal{S}|}$ and any $\mathbf{p} \in \Delta^{|\mathcal{S}|}$, define a cost function $\mathcal{C}(\mathbf{g}, \mathbf{p}) \triangleq -\sum_{i=1}^{|\mathcal{S}|} g_i \log p_i$, *i.e.*, a generalized cross-entropy function with the first argument not normalized to a probability vector. Note $\mathcal{C}(\mathbf{g}, \mathbf{p})$ is linear in $\mathbf{g}$ and convex in $\mathbf{p}$. Now, following lemma shows another way of obtaining DUDE.

**Lemma 1** *Define* $\mathbf{L}_{\text{new}} \triangleq -\mathbf{L} + L_{\max} \mathbf{1} \mathbf{1}^\top$ *in which* $L_{\max} \triangleq \max_{z,s} \mathbf{L}(z, s)$, *the maximum element of* $\mathbf{L}$. *Using the cost function* $\mathcal{C}(\cdot, \cdot)$ *defined above, for each* $\mathbf{c} \in \mathbf{C}_k$, *let us define*

$$\mathbf{p}^*(\mathbf{c}) \triangleq \arg\min_{\mathbf{p} \in \Delta^{|\mathcal{S}|}} \sum_{\{i : \mathbf{c}_i = \mathbf{c}\}} \mathcal{C}\big( \mathbf{L}_{\text{new}}^\top \mathbb{1}_{z_i}, \mathbf{p} \big).$$

*Then, we have* $s_{k,\text{DUDE}}(\mathbf{c}, \cdot) = \arg\max_s \mathbf{p}^*(\mathbf{c})_s$.

*Proof:* The proof of lemma is given in the Supplementary Material. ∎

## 4.2  Neural DUDE

The main idea for Neural DUDE is to use a *single* neural network to learn the $k$-th order slinding window denoising rule for all $\mathbf{c}$'s. Namely, we define $\mathbf{p}(\mathbf{w}, \cdot) : \mathcal{Z}^{2k} \to \Delta^{|\mathcal{S}|}$ as a feed-forward neural network that takes the context vector $\mathbf{c} \in \mathbf{C}_k$ as input and outputs a probability vector on $\Delta^{|\mathcal{S}|}$. We let $\mathbf{w}$ stand for all the parameters in the network. The network architecture of $\mathbf{p}(\mathbf{w}, \cdot)$ has the softmax output layer, and it is analogous to that used for the multi-class classification. Thus, when the parameters are properly learned, we expect that $\mathbf{p}(\mathbf{w}, \mathbf{c}_i)$ will give predictions on which single-symbol denoiser to apply at location $i$ with the context $\mathbf{c}_i$.

### 4.2.1  Learning

When not resorting to the supervised learning framework, learning the network parameters $\mathbf{w}$ is not straightforward as mentioned in the Introduction. However, inspired by Lemma 1, we define the objective function to minimize for learning $\mathbf{w}$ as

$$\mathcal{L}(\mathbf{w}, z^n) \quad \triangleq \quad \frac{1}{n} \sum_{i=1}^{n} \mathcal{C}\left(\mathbf{L}_{\text{new}}^{\top} \mathbb{1}_{z_i}, \mathbf{p}(\mathbf{w}, \mathbf{c}_i)\right), \tag{7}$$

which resembles the widely used cross-entropy objective function in supervised multi-class classification. Namely, in (7), $\{(\mathbf{c}_i, \mathbf{L}_{\text{new}}^{\top} \mathbb{1}_{z_i})\}_{i=1}^{n}$, which solely depends on the noisy sequence $z^n$, can be analogously thought of as the input-label pairs in supervised learning. (Note for $i \leq k$ and $i \geq n - k$, dummy variables are padded for obtaining $\mathbf{c}_i$.) But, unlike classification, in which the ground-truth label is given as a one-hot vector, we treat $\mathbf{L}_{\text{new}}^{\top} \mathbb{1}_{z_i} \in \mathbb{R}_{+}^{|\mathcal{S}|}$ as a target "pseudo-label" on $\mathcal{S}$.

Once the objective function is set as in (7), we can then use the widely used optimization techniques, namely, the back-propagation and Stochastic Gradient Descent (SGD)-based methods, for learning the parameters $\mathbf{w}$. In fact, most of the well-known improvements to the SGD method, such as the momentum [14], mini-batch SGD, and several others [15, 16], can be all used for learning $\mathbf{w}$. Note that there is no notion of *generalization* in our setting, since the goal of denoising is to simply achieve as small average loss as possible for the given noisy sequence $z^n$, rather than performing well on the separate unseen test data. Hence, we do not use any regularization techniques such as dropout in our learning, but simply try to minimize the objective function.

### 4.2.2  Denoising

After sufficient iterations of weight updates, the objective function (7) will converge, and we will denote the converged parameters as $\mathbf{w}^*$. The Neural DUDE algorithm then applies the resulting network $\mathbf{p}(\mathbf{w}^*, \cdot)$ to the exact same noisy sequence $z^n$ used for learning to denoise. Namely, for each $\mathbf{c} \in \mathbf{C}_k$, we obtain a single-symbol denoiser

$$s_{k,\texttt{Neural DUDE}}(\mathbf{c}, \cdot) = \arg \max_s \mathbf{p}(\mathbf{w}^*, \mathbf{c})_s \tag{8}$$

and the reconstruction at location $i$ by $\hat{X}_{i,\texttt{DUDE}}(z^n) = s_{k,\texttt{Neural DUDE}}(\mathbf{c}_i, z_i)$.

From the objective function (7) and the definition (8), it is apparent that Neural DUDE does share information across different contexts since $\mathbf{w}^*$ is learnt from all data and shared across all contexts. Such property enables Neural DUDE to robustly run with much larger $k$'s than DUDE without running into the data sparsity problem. As shown in the experimental section, Neural DUDE with large $k$ can significantly improve the denoising performance compared to DUDE. Furthermore, in the experimental section, we show that the concentration

$$\frac{1}{n} \sum_{i=1}^{n} \mathbf{L}(Z_i, s_{k,\texttt{Neural DUDE}}(\mathbf{c}_i, \cdot)) \approx \frac{1}{n} \sum_{i=1}^{n} \Lambda(x_i, s_{k,\texttt{Neural DUDE}}(\mathbf{c}_i, Z_i)) \tag{9}$$

holds with high probability even for very large $k$'s, whereas such concentration quickly breaks for DUDE as $k$ grows. While deferring the analyses on why such concentration always holds to the future work, we can use the property to provide a systematic mechanism for choosing the best context size $k$ for Neural DUDE - simply choose $k^* = \arg \min_k \frac{1}{n} \sum_{i=1}^{n} \mathbf{L}(Z_i, s_{k,\texttt{Neural DUDE}}(\mathbf{c}_i, \cdot))$. As shown in the experiments, such choice of $k$ for Neural DUDE gives an excellent denoising performace.

Algorithm 1 summarizes the Neural DUDE algorithm.

**Algorithm 1** Neural DUDE algorithm

---
**Input:** Noisy sequence $z^n$, $\mathbf{\Pi}$, $\mathbf{\Lambda}$, Maximum context size $k_{\max}$
**Output:** Denoised sequence $\hat{X}^n_{\texttt{Neural DUDE}} = \{\hat{X}_{i,\texttt{Neural DUDE}}(z^n)\}^n_{i=1}$
  Compute $\mathbf{L} = \mathbf{\Pi}^{-1}\boldsymbol{\rho}$ as in Section 3.1 and $\mathbf{L}_{\text{new}}$ as in Lemma 1
  **for** $k = 1, \dots, k_{\max}$ **do**
    Initialize $\mathbf{p}(\mathbf{w}, \cdot)$ with input dimension $2k|\mathcal{Z}|$ (using one-hot encoding of each noisy symbol)
    Obtain $\mathbf{w}^*_k$ minimizing $\mathcal{L}(\mathbf{w}, z^n)$ in (7) using SGD-like optimization method
    Obtain $s_{k,\texttt{Neural DUDE}}(\mathbf{c}, \cdot)$ for all $\mathbf{c} \in \mathbf{C}_k$ as in (8) using $\mathbf{w}^*_k$
    Compute $L_k \triangleq \frac{1}{n}\sum^n_{i=1}\mathbf{L}(z_i, s_{k,\texttt{Neural DUDE}}(\mathbf{c}_i, \cdot))$
  **end for**
  Get $k^* = \arg\min_k L_k$ and obtain $\hat{X}_{i,\texttt{Neural DUDE}}(z^n) = s_{k^*,\texttt{Neural DUDE}}(\mathbf{c}_i, z_i)$ for $i = 1, \dots, n$

---

*Remark:* We note that using the cost function in (7) is important. That is, if we use a simpler objective like (5), $\frac{1}{n}\sum^n_{i=1}(\mathbf{L}^\top\mathbb{1}_{z_i})^\top\mathbf{p}(\mathbf{w}, \mathbf{c}_i)$, it becomes highly non-convex in $\mathbf{w}$, and the solution $\mathbf{w}^*$ becomes very unstable. Moreover, using $\mathbf{L}_{\text{new}}$ instead of $\mathbf{L}$ in the cost function is important as well, since it guarantees to have the cost function $\mathcal{C}(\cdot, \cdot)$ always convex in the second argument.

# 5 Experimental results

In this section, we show the denoising results of Neural DUDE for the synthetic binary data, real binary images, and real Oxford Nanopore MinION DNA sequence data. All of our experiments were done with Python 2.7 and Keras package (http://keras.io) with Theano [17] backend.

## 5.1 Synthetic binary data

We first experimented with a simple synthetic binary data to highlight the core strength of Neural DUDE. That is, we assume $\mathcal{X} = \mathcal{Z} = \hat{\mathcal{X}} = \{0, 1\}$ and $\mathbf{\Pi}$ is a binary symmetric channel (BSC) with crossover probability $\delta = 0.1$. We set $\mathbf{\Lambda}$ as the Hamming loss. We generated the clean binary

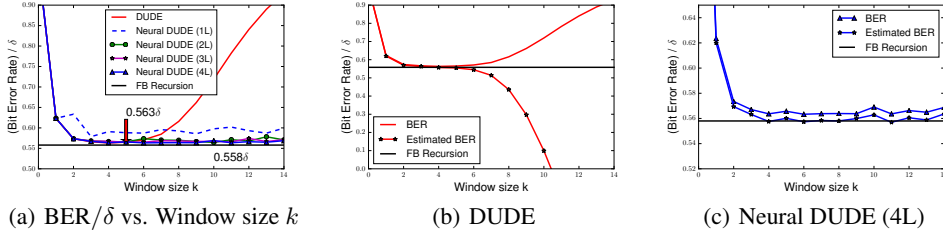

(a) BER/$\delta$ vs. Window size $k$      (b) DUDE      (c) Neural DUDE (4L)

Figure 1: Denoising results of DUDE and Neural DUDE for the synthetic binary data with $n = 10^6$.

sequence $x^n$ of length $n = 10^6$ from a binary symmentric Markov chain (BSMC) with transition probability $\alpha = 0.1$. The noise-corrupted sequence $z^n$ is generated by passing $x^n$ through $\mathbf{\Pi}$. Since we use the Hamming loss, the average loss of a denoiser $\hat{X}^n$, $\frac{1}{n}\sum^n_{i=1}\Lambda(x_i, \hat{X}_i(z^n))$, is equal to the bit error rate (BER). Note that in this setting, the noisy sequence $z^n$ is a hidden Markov process. Therefore, when the stochastic model of the clean sequence is exactly known to the denoiser, the Viterbi-like Forward-Backward (FB) recursion algorithm can attain the *optimum* BER.

Figure 1 shows the denoising results of DUDE and Neural DUDE, which do not know anything about the characteristics of the clean sequence $x^n$. For DUDE, the window size $k$ is the single hyperparameter to choose. For Neural DUDE, we used the feed-forward fully connected neural networks for $\mathbf{p}(\mathbf{w}, \cdot)$ and varied the depth of the network between $1 \sim 4$ while also varying $k$. Neural DUDE(1L) corresponds to the simple linear softmax regression model. For deeper models, we used 40 hidden nodes in each layer with Rectified Linear Unit (ReLU) activations. We used Adam [16] with default setting in Keras as an optimizer to minimize (7). We used the mini-batch size of 100 and ran 10 epochs for learning. The performance of Neural DUDE was robust to the initializtion of the parameters $\mathbf{w}$.

Figure 1(a) shows the BERs of DUDE and Neural DUDE with respect to varying $k$. Firstly, we see that minimum BERs of both DUDE and Neural DUDE(4L), *i.e.*, $0.563\delta$ with $k = 5$, get very close to the optimum BER ($0.558\delta$) obtained by the Forward-Backward (FB) recursion. Secondly, we observe that Neural DUDE quickly approaches the optimum BER as we increase the depth of the network. This shows that as the descriminative power of the model increases with the depth of the network, $\mathbf{p}(\mathbf{w}, \cdot)$ can successfully learn the denoising rule for each context $\mathbf{c}$ with a shared parameter $\mathbf{w}$. Thirdly, we clearly see that in contrast to the performance of DUDE being sensitive to $k$, that of Neural DUDE(4L) is robust to $k$ by sharing information across contexts. Such robustness with respect to $k$ is obviously a very desirable property in practice.

Figure 1(b) and Figure 1(c) plot the average estimated BER, $\frac{1}{n}\sum_{i=1}^{n} \mathbf{L}(Z_i, s_k(\mathbf{c}_i, \cdot))$, against the true BER for DUDE and Neural DUDE (4L), respectively, to show the concentration phenomenon described in (9). From the figures, we can see that while the estimated BER drastically diverges from true BER for DUDE as $k$ increases, it strongly concentrates on true BER for Neural DUDE (4L) for all $k$. This result suggests the concrete rule for selecting the best $k$ described in Algorithm 1. Such rule is used for the experiments using real data in the following subsections.

## 5.2   Real binary image denoising

In this section, we experiment with real, binary image data. The settings of $\mathbf{\Pi}$ and $\mathbf{\Lambda}$ are identical to Section 5.1, while the clean sequence was generated by converting image to a 1-D sequence via raster scanning. We tested with 5 representative binary images with various textual characteristics: Einstein, Lena, Barbara, Cameraman, and scanned Shannon paper. Einstein and Shannon images had the resolution of $256 \times 256$ and the rest had $512 \times 512$. For Neural DUDE, we tested with 4 layer model with 40 hidden nodes with ReLU activations in each layer.

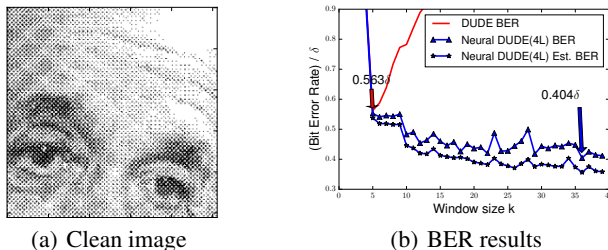

(a) Clean image          (b) BER results

Figure 2: Einstein image($256 \times 256$) denoising results with $\delta = 0.1$.

Figure 2(b) shows the result of denoising Einstein image in Figure 2(a) for $\delta = 0.1$. We see that the BER of Neural DUDE(4L) continues to drop as we increase $k$, whereas DUDE quickly fails to denoise for larger $k$'s. Furthermore, we observe that the estimated BER of Neural DUDE(4L) again strongly correlates with the true BER. Note that when $k = 36$, we have $2^{72}$ possible different contexts, which are much more than the number of pixels, $2^{16}(256 \times 256)$. However, we see that Neural DUDE can still learn a good denoising rule from such many different contexts by aggregating information from similar contexts.

| $\delta$ | Schemes | Einstein | Lena | Barbara | Cameraman | Shannon |
|---|---|---|---|---|---|---|
| | DUDE | 0.578 (5) | 0.494 (6) | 0.492 (5) | 0.298 (6) | 0.498 (5) |
| 0.15 | Neural DUDE | **0.384 (38)** | **0.405 (38)** | **0.448 (33)** | **0.264 (39)** | **0.410 (38)** |
| | Improvement | **33.6%** | **18.0%** | **9.0%** | **11.5%** | **17.7%** |
| | DUDE | 0.563 (5) | 0.495 (6) | 0.506 (6) | 0.310 (5) | 0.475 (5) |
| 0.1 | Neural DUDE | **0.404 (36)** | **0.403 (38)** | **0.457 (27)** | **0.268 (35)** | **0.402 (35)** |
| | Improvement | **28.2%** | **18.6%** | **9.7%** | **13.6%** | **15.4%** |

Table 1: BER results for binary images. Each number represents the relative BER compared to $\delta$ and the "Improvement" stands for the relative BER improvement of Neural DUDE(4L) over DUDE. The numbers inside parentheses are the $k$ values achieving the result.

Table 1 summarizes the denoising results on six binary images for $\delta = 0.1, 0.15$. We see that Neural DUDE always significantly outperforms DUDE using much larger context size $k$. We believe this is a

significant result since DUDE is shown to outperform many state-of-the-art sliding window denoisers in practice such as median filters [5, 1]. Furthermore, following DUDE's extension to grayscale image denoising [2], the result gives strong motivation for extending Neural DUDE to grayscale image denoising.

### 5.3   Nanopore DNA sequence denoising

We now go beyond binary data and apply Neural DUDE to DNA sequence denoising. As surveyed in [9], denoising DNA sequences is becoming increasingly important as the sequencing devices are getting cheaper, but injecting more noise than before. For our experiment, we used simulated MinION Nanopore reads, which were generated as follows; we obtained 16S rDNA reference sequences for 20 species [18] and randomly generated noiseless template reads from them. The number of reads and read length for each species were set as identical to those of real MinION Nanopore reads [18]. Then, based on $\Pi$ of MinION Nanopore sequencer (Figure 3(a)) obtained in [19] (with 20.375% average error rate), we induced substitution errors to the reads and obtained the corresponding noisy reads. Note that we are only considering substitution errors, while there also exist insertion/deletion errors in real Nanopore sequenced data. The reason is that substitution errors can be directly handled by DUDE and Neural DUDE, so we focus on quantitatively evaluating the performance on those errors. We sequentially merged 2,372 reads from 20 species and formed 1-D sequence of 2,469,111 base pairs long. We used two Neural DUDE (4L) models with 40 and 80 hidden nodes in each layer, and denoted as (40-40-40) and (80-80-80), respectively.

$$\Pi = \begin{bmatrix} 0.8122 & 0.0034 & 0.0894 & 0.0950 \\ 0.0096 & 0.8237 & 0.0808 & 0.0859 \\ 0.1066 & 0.0436 & 0.7774 & 0.0724 \\ 0.0704 & 0.0690 & 0.0889 & 0.7717 \end{bmatrix}$$

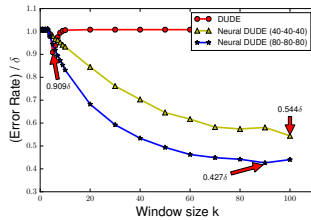

(a) $\Pi$ for nanopore sequencer                    (b) BER results

Figure 3: Nanopore DNA sequence denoising results.

Figure 3(b) shows the denoising results. We observe that Neural DUDE with large $k$'s (around $k = 100$) can achieve less than half of the error rate of DUDE. Furthermore, as the complexity of model increases, the performance of Neural DUDE gets significantly better. We could not find a comparable baseline scheme, since most of nanopore error correction tool, *e.g.*, Nanocorr [20], did not produce read-by-read correction sequence, but returns downstream analyses results after denoising. Coral [21], which gives read-by-read denoising result for Illumina data, completely failed for the nanopore data. Given that DUDE ourperforms state-of-the-art schemes, including Coral, for Illumina sequenced data as shown in [3], we expect the improvement of Neural DUDE over DUDE could translate into fruitful downstream analyses gain for nanopore data.

## 6   Concluding remark and future work

We showed Neural DUDE significantly improves upon DUDE and has a systematic mechanism for choosing the best $k$. There are several future research directions. First, we plan to do thorough experiments on DNA sequence denoising and quantify the impact of Neural DUDE in the downstream analysis. Second, we plan to give theoretical analyses on the concentration (9) and justify the derived $k$ selection rule. Third, extending the framework to deal with continuous-valued signal and finding connection with SURE principle [22] would be fruitful. Finally, applying recurrent neural networks (RNN) in place of DNNs could be another promising direction.

**Acknowledgments**

T. Moon was supported by DGIST Faculty Start-up Fund (2016010060) and Basic Science Research Program through the National Research Foundation of Korea (2016R1C1B2012170), both funded by Ministry of Science, ICT and Future Planning. S. Min, B. Lee, and S. Yoon were supported in part by Brain Korea 21 Plus Project (SNU ECE) in 2016.

## Footnotes

[1]We use uppercase letter $Z$ to stress it is a random variable

[2]For general case in which $\mathbf{\Pi}$ is not a square matrix, $\mathbf{\Pi}^{-1}$ can be replaced with the right inverse of $\mathbf{\Pi}$.

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
