[Supplementary Material]

# Supplementary Material for Neural Universal Discrete Denoiser

**Taesup Moon**
DGIST
Daegu, Korea 42988
tsmoon@dgist.ac.kr

**Seonwoo Min, Byunghan Lee, Sungroh Yoon**
Seoul National University
Seoul, Korea 08826
{mswzeus, styxkr, sryoon}@snu.ac.kr

## 1 Proof of Lemma 1

**Lemma 1** *Define* $\mathbf{L}_{\text{new}} \triangleq -\mathbf{L} + L_{\max}\mathbf{1}\mathbf{1}^{\top}$ *in which* $L_{\max} \triangleq \max_{z,s} \mathbf{L}(z,s)$, *the maximum element of* $\mathbf{L}$. *Using the cost function* $\mathcal{C}(\cdot,\cdot)$ *defined above, for each* $\mathbf{c} \in \mathbf{C}_k$, *let us define*

$$\mathbf{p}^*(\mathbf{c}) \triangleq \arg \min_{\mathbf{p} \in \Delta^{|\mathcal{S}|}} \sum_{\{i:\mathbf{c}_i=\mathbf{c}\}} \mathcal{C}\big(\mathbf{L}_{new}^{\top}\mathbb{1}_{z_i}, \mathbf{p}\big).$$

*Then, we have* $s_{k,\text{DUDE}}(\mathbf{c},\cdot) = \arg \max_s \mathbf{p}^*(\mathbf{c})_s$.

*Proof:* Recalling

$$\hat{\mathbf{p}}(\mathbf{c}) \triangleq \arg \min_{\mathbf{p} \in \Delta^{|\mathcal{S}|}} \Big( \sum_{\{i:\mathbf{c}_i=\mathbf{c}\}} \mathbb{1}_{z_i}^{\top}\mathbf{L} \Big)\mathbf{p}, \tag{1}$$

we derive

$$\hat{\mathbf{p}}(\mathbf{c}) = \arg \max_{\mathbf{p} \in \Delta^{|\mathcal{S}|}} \Big( \sum_{\{i:\mathbf{c}_i=\mathbf{c}\}} \mathbb{1}_{z_i}^{\top}\underbrace{(-\mathbf{L} + L_{\max}\mathbf{1}\mathbf{1}^{\top})}_{=\mathbf{L}_{\text{new}}} \Big)\mathbf{p} = \arg \max_{\mathbf{p} \in \Delta^{|\mathcal{S}|}} \Big( \sum_{\{i:\mathbf{c}_i=\mathbf{c}\}} \mathbf{L}_{\text{new}}^{\top}\mathbb{1}_{z_i} \Big)^{\top}\mathbf{p} \tag{2}$$

in which the first equality follows from flipping the sign of $\mathbf{L}$ and the fact that $\arg \max$ does not change by adding a constant to the objective. Furthermore, since $\mathcal{C}(\cdot,\cdot)$ is linear in the first argument,

$$\mathbf{p}^*(\mathbf{c}) = \arg \min_{\mathbf{p} \in \Delta^{|\mathcal{S}|}} \sum_{i \in \{\mathbf{c}_i=\mathbf{c}\}} \mathcal{C}\big(\mathbf{L}_{\text{new}}^{\top}\mathbb{1}_{z_i}, \mathbf{p}\big) = \arg \min_{\mathbf{p} \in \Delta^{|\mathcal{S}|}} \mathcal{C}\Big( \sum_{i \in \{\mathbf{c}_i=\mathbf{c}\}} \mathbf{L}_{\text{new}}^{\top}\mathbb{1}_{z_i}, \mathbf{p} \Big). \tag{3}$$

Now, from comparing (2) and (3), and from the fact that $\sum_{i \in \{\mathbf{c}_i=\mathbf{c}\}} \mathbf{L}_{\text{new}}^{\top}\mathbb{1}_{z_i} \in \mathbb{R}_+^{|\mathcal{S}|}$, we can show that

$$\arg \max_s \hat{\mathbf{p}}(\mathbf{c})_s = \arg \max_s \mathbf{p}^*(\mathbf{c})_s = \arg \max_s (\sum_{i \in \{\mathbf{c}_i=\mathbf{c}\}} \mathbf{L}_{\text{new}}^{\top}\mathbb{1}_{z_i})_s$$

by considering Lagrangian of (3) and applying KKT condition. That is, $\mathbf{p}^*(\mathbf{c})$ no longer is on one of the vertex of $\Delta^{|\mathcal{S}|}$, but still puts the maximum probability mass on the vertex $\hat{\mathbf{p}}(\mathbf{c})$. Since $s_{k,\text{DUDE}}(\mathbf{c},\cdot) = \arg \max_s \hat{\mathbf{p}}(\mathbf{c})_s$ as shown in the previous section, the proof is done. ∎