[Reviews · NeurIPS 2016]

Reviewer 1

Summary

This paper presents a framework for learning a sliding window denoiser. In contrast to a previous approach where the map for each context is adjusted independently, here the idea is to use a neural network which can share parameters for different contexts and allows to use larger sliding windows. Training is based on certain pseudo labels derived from the input stream statistics.

Qualitative Assessment

* The idea of using a single neural network to compute the demonising rule for various different contexts appears sensible but not particularly deep. * Section 5.2 is testing 5 individual images. This does not seem very representative. * eq. (5). Here it seems that the argument z_i should be included * lines 230, 266. typos `restricted', `following' ** I have adjusted my novelty rating after the discussion. I agree that the development of pseudo labels and the objective function has enough novelty and is quite interesting.

Confidence in this Review

2-Confident (read it all; understood it all reasonably well)


Reviewer 2

Summary

This paper proposes an interesting extension of the DUDE denoising algorithm through the use of deep neural networks (Neural DUDE). The paper uses the idea of weight sharing in a single neural networks to allow for a better estimation of a denoiser, it proposes an empirical rule to select a denoising window, and it compares the performances of Neural DUDE against DUDE.

Qualitative Assessment

The paper is very interesting in its idea of using neural networks to learn a DUDE module. The discussion is very neat and rigorous, even if, at times, a little bit harder to follow without carefully checking the references. If possible, I think it would be interesting to invest a little bit more explanation information sharing through weight sharing (line 150-151). Conceptually, I have been a little bit troubled by the fact that the neural network is not required to generalize (line 184); can this still be called 'learning'? Few minor typos: computatioanlly -> computationally (line 17); back-propagaion -> back-propagation (line 181); estmated -> estimated; are -> is (line 271). Slightly confusing footnoting: star is easily mistaken for mathematical notation, especially after a symbol (line 113); where is the footnote for line 177?

Confidence in this Review

1-Less confident (might not have understood significant parts)


Reviewer 3

Summary

The paper presents a novel denoiser for long discrete sequences corrupted by iid noise. The denoiser is provides with a noise model (symbol substitution probabilities) and cost of symbols substitution, but is not provided with clean data. The denoiser is composed of an neural network that predicts "pseudo-labels", which are in-fact a distribution over single-symbol denoisers (conditioned on input context).

Qualitative Assessment

The paper is clearly written, well motivated, and presents an interesting and novel technique, supported by convincing experiments. I find the loss function used very interesting, and the connections with previous work (DUDE) shed light on how it works. The following remarks are only to improve the state of the paper: - title: The technique is called Neural-DUDE (Discrete Universal DEnoiser) so a better title would be: "Neural Discrete Universal Denoiser) - Section 2.2, it would help reading if k was (re)-defined there (window size) - line 99: instead of "Introduction" it should be "the introduction" - equation 3: shouldn't the expectation be over z ? (the random variable). - in section 4.2.1 the authors use "I" for the indicator function, which is inconsistent with the rest of the paper (e.g. lemma 1). - line 235: "gets" instead of "get. - line 236, "FB recursion" should be spelled out. - line 266: "folloiwng" -> "following" - line 284: A larger line spacing is needed. - line 286: consider using "a comparable" instead of "right"

Confidence in this Review

2-Confident (read it all; understood it all reasonably well)


Reviewer 4

Summary

This paper proposed a neural universal discrete denoiser. It is an extension of the universal discrete denoiser (DUDE) based on a deep neural network. The DNN can be trained using the so-called "pseudo-labels" under a novel objective function. Experimental results show some significant improvements over the conventional DUDE.

Qualitative Assessment

The core of the work is to use a DNN as an estimator of the empirical distribution of symbols given the context. The authors also designed a novel objective function that does not require labeled data. Not surprisingly, the DNN used in the presented Neural DUDE turns out to be a pretty good estimator with shared parameters. Significant improvements were observed. This is an elegant piece of work, an interesting extension of the convention DUDE. I don't have problems with the Neural DUDE per se. But I have following concerns or comments. 1. Both DUDE and Neural DUDE will see a serious data sparsity problem when the alphabet becomes large. So far, a lot of application examples of DUDE are limited to very small alphabets, e.g. black-white images (|X|=2) and DNA sequencing (|X|=4). Neural DUDE uses a DNN with parameter sharing can mitigate it to some degree but I guess it will quickly run into problems when the context is long with a big alphabet. For example, the authors mentioned literature [4] and [22] in the introduction as related work using supervised training. But I wonder how the neural DUDE will perform when dealing with grayscale images as used in [4] and [22] rather than the black-white binary images used in this work? The alphabets are much bigger in [4] and [22]. So, although it is elegant in theory, it is hard to see it makes an impact to the real image denoising applications. 2. Another concern about the experiments is that the authors assume the channel is known. In other words, the channel transition matrix is known before hand. This is not a very realistic assumption in real world scenarios. Most cases, you don't know the channel and you have to estimate it. This is another crucial issue for the neural DUDE to make a real impact. How will the estimated channel transition matrix affect the performance? When the alphabet is big, the estimation of the matrix and its inversion itself may pose a problem.

Confidence in this Review

2-Confident (read it all; understood it all reasonably well)


Reviewer 5

Summary

The paper presents a new framework of applying deep neural networks (DNN) to devise a universal discrete denoiser. Compared with the other approaches that utilize supervised learning for denoising, the new method do not require any additional training data.

Qualitative Assessment

Theoretical justification for the new algorithms should be further improved.

Confidence in this Review

1-Less confident (might not have understood significant parts)